# Green-KGC: Automated Semantic Lifting and Schema Inference for Open Government Data via Traditional Machine Learning

Michel G. Recondo[1,*,†], M.Sc.

[1]*Universidade Tecnolóica Federal do Paraná (UTFPR), Av. Sete de Setembro, 3165, Curitiba PR Brasil*

## Abstract

The openness of government data in Latin America faces structural challenges regarding semantic interoperability. Most portals stagnate at level 3 of the 5-star open data scale, limiting knowledge discovery and perpetuating tabular silos. This paper proposes an automated, computationally efficient framework to elevate legacy tabular data to level 5 (Linked Open Data). Unlike resource-intensive Deep Learning approaches for semantic type detection, which impose prohibitive infrastructure costs on public institutions, this research investigates the efficacy of lightweight algorithms (e.g., Random Forest). By engineering statistical features from raw CSVs, the pipeline infers ontological classes to automate the generation of declarative mapping rules (RML), ensuring that Knowledge Graph Construction remains accessible, scalable, and sustainable for resource-constrained e-government infrastructures.

## Keywords

Knowledge Graph Construction, Semantic Lifting, Schema Inference, Open Government Data

## 1. Introduction

The paradigm of Open Government Data (OGD) has evolved significantly over the past decade, transitioning from a mere legal obligation for transparency into a strategic asset for evidence-based policymaking and innovation. However, unlocking the full potential of these datasets is currently hindered by a severe technical bottleneck: the pervasive lack of semantic interoperability. Empirical evidence indicates that the vast majority of government transparency portals, particularly in Latin America, operate as disconnected silos of raw files. Within the classic 5-star open data deployment scheme proposed by Berners-Lee [1], most public administrations have stagnated at the 3-star level, making massive volumes of information available in non-proprietary tabular formats, such as CSV. The necessary leap to the 5-star level—where data is fully structured as Linked Open Data (LOD) and integrated into global Knowledge Graphs—remains an elusive goal due to the historical complexity, cost, and slow pace of semantic materialization.

Transitioning from tabular silos to robust Knowledge Graphs requires a process known as semantic lifting. Traditionally, this process relies on domain experts manually crafting declarative mapping rules—using standards such as the RDF Mapping Language (RML) [2]—to dictate how raw columns should be translated into ontological triples. Given the continuous generation of data and the accumulation of historical records by public administrations, relying on manual curation is fundamentally unscalable. To overcome this bottleneck, the data integration community has increasingly turned towards automation, with recent state-of-the-art approaches heavily favouring Deep Learning architectures to perform semantic type detection and schema inference.

While these Deep Learning models, such as Sherlock [3] and its successors, exhibit remarkable accuracy in identifying semantic types across heterogeneous datasets, their application within the context of public administration reveals a critical misalignment. Government entities, especially at the municipal and regional levels, frequently operate with constrained budgets and heterogeneous IT

*Seventh International Workshop on Knowledge Graph Construction, May, 2026, Dubrovnik, Croatia*

*Corresponding author.

✉ michelrecondo@alunos.utfpr.edu.br (M. G. Recondo)

⬡ 0009-0005-2083-9983 (M. G. Recondo)

infrastructures. The requirement for high-performance computing clusters and power-intensive GPUs to continuously train and run deep neural networks makes the adoption of such advanced semantic technologies financially prohibitive and environmentally unsustainable. Consequently, the reliance on resource-intensive Artificial Intelligence deepens the technological divide rather than democratizing Knowledge Graph Construction (KGC).

In response to these challenges, this paper proposes an automated, computationally efficient framework for the semantic lifting of open government data, firmly rooted in the principles of Green AI. It is hypothesized that the intrinsic morphological and statistical properties of tabular government data allow for highly accurate schema inference without the reliance on massive neural networks. By extracting statistical features, character distributions, and information entropy metrics from raw CSV files, the proposed pipeline trains lightweight, traditional Machine Learning algorithms—such as Random Forest [4]—to predict ontological classes. Crucially, these predictions are then used to dynamically generate RML mapping scripts, automating the most labor-intensive phase of KGC. This framework aims to provide a scalable solution to the semantic interoperability crisis, ensuring that the construction of public Knowledge Graphs remains an accessible, sustainable, and resilient endeavor for resource-constrained e-government infrastructures.

## 2. The Semantic Bottleneck and Deep Learning Limitations

The automation of Knowledge Graph Construction from tabular sources inherently depends on the accurate resolution of the semantic typing problem. To dynamically generate mapping rules, a system must first infer the ontological meaning of a given tabular column—for instance, distinguishing whether a sequence of integers represents a geographic coordinate, a temporal constraint, or a financial budget. Historically, this semantic inference relied on rigid rule-based systems and regular expressions, which proved brittle and unscalable when confronted with the vast morphological heterogeneity of open government datasets.

Consequently, the state of the art in semantic type detection has rapidly shifted towards advanced probabilistic models and Deep Learning architectures. Frameworks such as Sherlock [3] represent the vanguard of this approach, employing deep neural networks trained on massive tabular corpora, such as VizNet, to classify over a thousand distinct semantic types. Subsequent extensions, such as Sato (Semantic Type Detection with Multi-column Context) [5], further improved classification precision by incorporating the global spatial context of the table into the neural architecture. While these deep models achieve remarkable performance benchmarks, their deployment introduces a severe structural paradox when transposed to the domain of public administration.

The primary limitation of Deep Learning approaches in e-government scenarios lies in their exorbitant computational and infrastructural demands. Retraining and executing inference on multi-layered neural networks requires substantial hardware acceleration, typically in the form of power-intensive Graphics Processing Units (GPUs). Within the context of Latin American municipalities, where IT infrastructures are highly heterogeneous and frequently resource-constrained, prioritizing computational efficiency and energy sustainability is not merely an ideological preference, but a strict non-functional requirement. Deep Learning models impose financial and operational barriers that threaten to make semantic interoperability an exclusive capability of well-funded central governments, thereby marginalizing local administrations and contradicting the foundational ethos of open data.

In contrast to the prevailing trend of utilizing massive neural networks, there is a notable scarcity of research investigating the efficacy of lightweight, traditional Machine Learning algorithms for the semantic lifting of tabular data. It is posited that the structural regularity and intrinsic statistical distribution of government datasets can be effectively captured through rigorous feature engineering. By extracting lightweight numerical metrics—such as global statistics, character distributions, and Shannon entropy—traditional classifiers like Random Forest [4] can be optimized to perform highly accurate schema inference. This computationally frugal approach avoids the overhead of massive language models, offering a sustainable, Green AI alternative for the automated generation of mapping

rules in environments where computational resources are strictly limited.

## 3. The Green-KGC Architecture: Automated Semantic Lifting

To operationalize the semantic lifting of tabular data at a governmental scale, this paper proposes an end-to-end, automated pipeline designed to bridge the gap between raw CSV ingestion and RDF triple materialization. The core of the Green-KGC architecture relies on extracting intrinsic statistical properties from the data to fuel a supervised, lightweight classifier. By shifting the computational burden from deep neural inference to rigorous feature engineering, the framework ensures a sustainable pathway for Knowledge Graph Construction. The pipeline is structured into three sequential phases: feature extraction, semantic inference, and automated mapping generation. As demonstrated in Figure 1, the pipeline is structured into three sequential phases: feature extraction, semantic inference, and automated mapping generation.

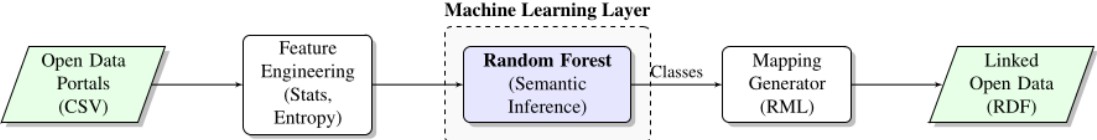

**Figure 1:** Architecture of the proposed Green-KGC framework for automated semantic lifting.

### 3.1. Tabular Feature Engineering

The success of traditional Machine Learning algorithms in semantic type detection is strictly dependent on the quality and discriminative power of the engineered features. Instead of processing raw textual embeddings—which is computationally expensive and typical of Deep Learning models—the proposed architecture extracts a deterministic set of meta-features for each column in the dataset. These features are grouped into three primary categories:

- **Global Statistics:** For numerical columns, statistical moments such as mean, variance, kurtosis, and skewness are calculated. These metrics are fundamental for differentiating semantic types that share the same primitive data type. For example, a column representing a "Year" will exhibit a significantly lower variance than a column representing a "Financial Budget", despite both being composed of numeric strings.
- **Character Distribution:** This category captures the morphological structure of the data by calculating the relative percentages of digits, alphabetic characters, and special symbols within a column. This distribution is highly effective for identifying standardized government formats, such as postal codes, procedural identification numbers, and dates, which possess rigid syntactic patterns.
- **Information Entropy and Uniqueness:** Evaluating the cardinality and the Shannon entropy of a column provides critical insights into its semantic nature. Categorical variables, such as "City District" or "Administrative Department", inherently present low entropy and a high recurrence of values. Conversely, columns containing primary keys, document numbers, or unique identifiers exhibit maximum entropy.

### 3.2. Semantic Inference via Lightweight Machine Learning

Once the feature vectors are extracted, they serve as the input for the semantic inference module. The proposed architecture employs a Random Forest classifier [4] to predict the ontological class (e.g., `schema:Person`, `xsd:date`) corresponding to each tabular column. Random Forest was selected due to its inherent robustness to outliers, its ability to model complex, non-linear interactions among

statistical features, and, most importantly, its exceptional computational efficiency during both the training and inference phases. This module effectively maps the morphological signature of a column to a standardized semantic vocabulary without necessitating hardware acceleration.

### 3.3. Automated Generation of Declarative Mappings

The defining contribution of the Green-KGC architecture to the Semantic Web community resides in its final transformation phase. While predicting semantic types is a critical intermediate step, the ultimate goal of Knowledge Graph Construction is materialization. To achieve this, the predictions generated by the Random Forest model are dynamically consumed by a mapping generator. This component autonomously translates the inferred semantic types into declarative mapping scripts formatted in the RDF Mapping Language (RML) [2].

By programmatically writing RML rules, the architecture eliminates the need for human domain experts to manually draft mappings for thousands of heterogeneous tabular files. These automatically generated scripts are then executed by standard RML processors to materialize the Linked Open Data triples. Consequently, the pipeline provides a fully automated, Green AI solution that transforms isolated, 3-star open data silos into interconnected, 5-star Knowledge Graphs with minimal computational overhead.

## 4. Feasibility Context: The Curitiba Open Data Ecosystem

To contextualize the practical urgency of automated semantic lifting, this research utilizes the open data ecosystem of Curitiba as a primary feasibility testbed [6]. The Curitiba Open Data Portal represents a quintessential scenario of governmental Big Data, comprising a historical depth of over a decade and amassing 3.35 terabytes of information distributed across more than 147,000 files. A recent empirical characterization of this repository revealed a striking technical paradox: while the portal exhibits an exceptionally high structural consistency—with 99.32% of its files strictly conforming to standard CSV formatting specifications (RFC 4180)—it simultaneously suffers from a severe lack of semantic standardization.

The metadata within this ecosystem predominantly relies on proprietary, localized schema conventions, such as headers formatted in UPPERCASE_UNDERSCORE, rather than globally recognized semantic vocabularies like DCAT. Furthermore, the datasets present a global sparsity rate of 59.72%, meaning that more than half of the tabular cells are completely empty. This scenario robustly validates the premise that the sheer availability of structurally sound CSV files does not automatically translate to semantic interoperability. The overwhelming volume of historical data, coupled with high sparsity and idiosyncratic schemas, renders manual ontological mapping functionally impossible. Consequently, this environment perfectly justifies the imperative application of computationally efficient Machine Learning techniques to perform automated schema inference and semantic population at scale.

## 5. Final Remarks and Future Work

This position paper advocates for a fundamental paradigm shift in the construction of governmental Knowledge Graphs, arguing that the transition from isolated tabular data (3-star) to Linked Open Data (5-star) must be computationally sustainable. By eschewing resource-intensive Deep Learning architectures in favor of traditional, lightweight Machine Learning algorithms driven by rigorous tabular feature engineering, the proposed framework offers a viable path to democratize advanced semantic technologies for resource-constrained public administrations. Furthermore, the dynamic translation of statistical predictions into declarative RML scripts represents a crucial step toward fully automated, end-to-end data integration pipelines that do not rely on constant human intervention.

Future work will focus on the empirical implementation and rigorous evaluation of the proposed Random Forest schema inference module using the Curitiba dataset. To accurately assess the model's

performance, the evaluation methodology will deliberately avoid simple global accuracy metrics, which can be highly misleading in governmental datasets dominated by generic string classes. Instead, the validation will rely on the Macro-F1 Score [7]. This metric calculates the harmonic mean between precision and recall by assigning equal weight to all classes, ensuring that the model's efficacy in detecting rare, high-value semantic types is not masked by the overwhelming prevalence of majority classes. Finally, upon validation, the inference pipeline is planned to be encapsulated within Docker containers and integrated as an API microservice into the CKAN platform, bridging the critical gap between theoretical schema inference and practical e-government deployment.

## Declaration on Generative AI

During the preparation of this work, the author used Google Gemini 3 in order to: Grammar and spelling check, text translation, and peer review simulaton. Further, the author used the tool for figure 1 in order to: Generate images. After using these tool, the author reviewed and edited the content as needed and take full responsibility for the publication's content.

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
