# OpenReview forum: "Green-KGC: Automated Semantic Lifting and Schema Inference for Open Government Data via Traditional Machine Learning"
_eswc-conferences.org/ESWC/2026/Workshop/KGCW — Submitted to KGCW 2026_

### Official Review · ~Hannes_Voigt1 · 2026-03-24
**A vision for an engineering project rather than a research vision.**

**Rating:** 4
**Confidence:** 3

**Review:**

This is a vision paper that argues for the semantic lifting of government data using lightweight ML tools (specifically Random Forest) instead of deep learning. The main argument is that deep learning techniques are not affordable for public administrations at the municipal and regional levels, which are the providers of such data. The vision appears reasonable in addressing a practical problem. However, it is not particularly novel or groundbreaking. The proposal would be more convincing if it included preliminary empirical evidence demonstrating that Random Forest can achieve suitable quality. Furthermore, the paper does not pose any explicit research questions. Overall, it reads more like a proposal seeking funding for an engineering project than a research vision.

Detailed remarks:

- Sections 1 and 2 are fairly redundant, as they both argue why deep learning is not suitable.

- The argument against deep learning seems to focus only on approaches that involve training specialized models. What about using existing foundation models? There are open models available, and inference is not as computationally intensive as training. At the very least, this aspect should be acknowledged.

- "The pipeline is structured into three sequential phases: feature extraction, semantic inference, and automated mapping generation. As demonstrated in Figure 1,
the pipeline is structured into three sequential phases: feature extraction, semantic inference, and automated mapping generation.": These sentences repeat the same content. One of them would suffice.

- The second paragraph presents an argument that is unlikely to be new to the audience of this venue; it largely preaches to the choir.

- "Further, the author used the tool for figure 1 in order to: Generate images": Is this intended to indicate that Figure 1 was generated using AI? If so, that is acceptable, but the colon (”:”) in the middle of the sentence seems out of place.

- Katrin Braunschweig’s work on web tables may be of interest to the author: https://nbn-resolving.org/urn:nbn:de:bsz:14-qucosa-184502 and https://scholar.google.de/citations?user=7U56AcAAAAAJ

---

### Official Review · ~Xuemin_Duan1 · 2026-04-04
**A position paper built on an already-explored baseline with an unclear performance ceiling**

**Rating:** 3
**Confidence:** 4

**Review:**

This position paper addresses the lack of semantic interoperability in open government data. The proposed framework, Green-KGC, argues that existing deep learning approaches for semantic type detection are too resource-intensive for public institutions, and instead proposes using lightweight traditional machine learning (specifically Random Forest) to conduct semantic table annotation and finally generate RML mapping. However, the proposed method's position relative to existing work is unclear and insufficiently supported by the literature. And the proposed method has a limited performance ceiling based on what is already known from prior work, and there is currently no basis to expect it to achieve practically useful performance.

**Importance and Relevance**

The problem addressed in this paper has clear practical value. The lack of semantic interoperability is a well-known challenge in open data ecosystems, and exploring low-cost KG construction for resource-constrained local governments is a direction worth pursuing. Combining Green AI principles with declarative KGC, particularly automatic RML generation, is well aligned with the core topics of the KGCW.

**Novelty**

The novelty of this paper is a serious concern.
This paper positions "using traditional ML instead of deep learning for semantic type detection" as the core contribution, but earlier work already explored traditional ML approaches. For example, Sherlock, which is the main reference work cited in this paper, already includes Random Forest as a baseline, using feature categories that closely match those proposed in this paper. Sherlock's results clearly show that the Random Forest baseline performs worse than its neural network on this task.
This paper needs more literature to support its novelty.

**Major Comments**
- The proposed method is not adequately positioned relative to existing work: The paper motivates its approach by arguing that DL methods are too resource-intensive for government institutions, and proposes a lightweight statistical feature-based approach as an alternative.  However, the paper does not cover enough existing work on feature-based and traditional ML methods for semantic type detection. The cited Sherlock already includes a Random Forest baseline trained on a feature set that closely overlaps with the one proposed here, covering global statistics and character distributions. Its results show that this baseline performs below the neural network. Beyond these, the paper does not discuss a large body of related work, such as the SemTab challenge series. Without covering this literature, it is not possible to identify what this paper contributes beyond what has already been studied, and the performance expectations of the proposed approach remain entirely unclear.

- The proposed method has a limited performance ceiling: This is the most critical technical concern. The paper proposes training a Random Forest on three categories of statistical features to predict ontological classes. Based on what is already known from prior work, this design has a very limited performance ceiling: (1) the proposed feature engineering seems does not mention the column headers; (2) Removing semantic representations is a hard restriction on performance; (3) Sherlock's baseline results already set the reference point for such methdoology; (4) High sparsity further reduces the reliability of statistical features.

- Semantic type inference for numerical columns is a much harder problem than acknowledged: The paper uses the example of a "Year" column versus a "Financial Budget" column having different variances to illustrate the effectiveness of statistical features. This example is overly optimistic. Inferring the semantic type of a column from its numerical values alone is nearly impossible in most cases, and typically requires contextual information from other text columns in the same table.

- Miss of target performance level: The paper's main selling point is lower resource consumption, but it never specifies what level of performance the proposed method is expected to achieve.

- The figure misses the ontology.

- How the training labels for Random Forest are obtained is not mentioned.

- The argumentation in Section 4 contains two logical gaps: (1) Sparsity and semantic interoperability are independent problems. The paper links the sparsity rate to the inability of CSV files to achieve semantic interoperability. However, the sparsity is a data quality issue, not a semantic interoperability issue, and the two should not be conflated. (2) The connection between UPPERCASE_UNDERSCORE naming and the absence of DCAT is unclear. The former is a local column naming convention, while the latter is a vocabulary. These operate at entirely different semantic levels, and the logical step from one to the other is not sufficiently explained.

**Minor Comment**
Section 3 has twice "feature extraction, semantic inference, and automated mapping generation".

This paper addresses a practically relevant problem, and the motivation to explore computationally sustainable approaches to government data semantic lifting is a direction worth pursuing. But it needs substantial revision based on the above comments.

---

### Official Review · ~Femke_Ongenae1 · 2026-04-06
**Would be better as a use case paper when (preliminary) results are included**

**Rating:** 4
**Confidence:** 4

**Review:**

This paper proposes a lightweight, Green AI pipeline for automated semantic lifting of open government data, relying on feature engineering and Random Forest classification instead of deep learning. The aim is to make semantic type inference and mapping generation computationally affordable for resource‑constrained public administrations.

The motivation is timely and relevant, particularly given current concerns around the sustainability and accessibility of Large Machine Learning models, i.e. used for Knowledge Graph Construction (KGC). However, several conceptual and methodological issues require clarification.

First, the paper argues that deep learning approaches are prohibitively expensive for public institutions. However, while training deep models is indeed compute‑intensive, inference is usually far less demanding, especially when using pre‑trained models, such as the referenced Sherlock model. It remains unclear why such pre‑trained models would not already suffice for most semantic typing tasks, why continuous training would be necessary, or why a more pragmatic combination of a pre‑trained model with lightweight post‑processing—possibly even a simple ML model—could not resolve remaining gaps. In many real-world scenarios, the remaining errors might even be small enough that a human‑in‑the‑loop strategy becomes the most straightforward and computationally efficient solution. A demonstration of how well these existing pre‑trained models perform on the Curitiba Open Data sets would therefore significantly strengthen the argument, as it would clarify the magnitude and nature of the remaining gaps and justify the need for alternative, less computationally intensive methods.

Second, the idea of generating mappings using simple machine learning models instead of deep learning has been stipulated before (I think the research on Karma has proposed simple supervised ML models for mapping generation). Therefore, a more thorough outline of existing state-of-the-art on the topic to position the work would be beneficial.

Third, the paper could benefit from better structuring. The justification for democratizing semantic lifting (in the Introduction) and the critique on deep learning (section 2) is very repetitive across these two sections, repeating the same arguments. It would be better to fuse these two sections into one. Additionally, Section 3 contains a minor repetition at the end of its introduction, i.e. "The pipeline is structured into three sequential phases: feature extraction, semantic inference, and automated mapping generation. As demonstrated in Figure 1, the pipeline is structured into three sequential phases: feature extraction, semantic inference, and automated mapping generation."

Overall, the paper does not fully read as a position paper, as the ideas presented are neither particularly novel nor sufficiently provocative to constitute a strong argumentative stance. Instead, it has the potential to be an interesting and valuable use‑case paper. However, to meet that standard, it would need to include at least some preliminary results, which are currently lacking.

---

### Decision · Program_Chairs · 2026-04-09

Reject